# A Pilot Study in Humans on the Urinary Tract Excretion of the FimH Inhibitor 1-Deoxymannose

**DOI:** 10.3390/antibiotics14050498

**Published:** 2025-05-13

**Authors:** Hiromi Hayashi, Naoto Miyazaki, Takuya Kawakami, Shusaku Izumi, Kazuhiro Yoshinaga

**Affiliations:** SUNUS Co., Ltd., 20 Nanei 3-Chome, Kagoshima City 891-0196, Kagoshima, Japan; rdkyoyo@sunus.co.jp (H.H.); nmiyazaki@sunus.co.jp (N.M.); tkawakami@sunus.co.jp (T.K.); syuu-izumi@sunus.co.jp (S.I.)

**Keywords:** 1-deoxymannose, 1,5-anhydro-D-mannitol, urinary tract infection, FimH

## Abstract

Background: FimH inhibitors are anticipated to serve as preventive therapeutics against urinary tract infections. Consequently, multiple inhibitors—predominantly D-mannose derivatives—have been synthesized, and their binding affinities (determined by dissociation coefficient; K_D_) to FimH have been examined in vitro. Nevertheless, the amounts of most of these synthetic compounds that reach the urinary tract after oral administration in humans have not been investigated. D-mannose (Man) and its analog, 1-Deoxymannose (DM), have already been reported to show K_D_ values against FimH. Therefore, this study aimed to estimate the post-oral ingestion of FimH inhibitory potentials of DM and Man in the urinary tract. Methods: Six participants were given single 1 g doses of DM and Man in a crossover test. The sample concentrations in urine were measured 8 h after ingestion. Results: DM levels increased rapidly after oral intake; contrarily, Man was detected in the urine before administration, with no notable increase post-ingestion. The peak concentration ranges of Man and DM in urine were 2.15–22.9 μg/mL and 665–57,804 μg/mL, respectively, which are 66.3–707 and 3600–31,200 times that of K_D,_ respectively. Conclusions: These results indicate that DM as a supplement is an orally active FimH inhibitor in the human urinary tract. Conversely, d-mannose is expected to exert comparatively milder inhibition.

## 1. Introduction

Urinary tract infection (UTI) is a disease stemming from the urinary system, such as the bladder, and is predominantly attributed to uropathogenic *Escherichia coli* (UPEC). The number of patients affected is second only to pneumonia, and sizeable medical expenses are incurred worldwide. Furthermore, even if cured, over 50% of patients experience recurrence within a year of recovery [1]. This is extremely high compared to the recurrence rate of pneumonia, which is 18.3%, and reduces patients’ quality of life [2]. Antibiotics are generally administered to treat this disease, but the overuse of antibiotics can result in the development of resistant bacteria, making antibiotics less effective. Occasionally, it is impossible to treat. As mentioned above, the appropriate use of antibiotics for the treatment of diseases is needed worldwide, and there is a need to develop alternatives to antibiotics for disease prevention. D-mannose (Man) is sold as a supplement to prevent urinary tract infections [3]. Orally ingested Man reportedly reaches the urinary tract and binds to the FimH protein of UPEC, inhibiting attachment to the mucosa [4]. However, a recent study concluded that Man was ineffective in preventing urinary tract infections [5]. Furthermore, in a human study, in which people who had experienced recurrent urinary tract infections ingested 2 g of Man, 12 showed no increase in urinary Man in the first hour after administration, while 7 showed an increase. Compared to before intake, the increase rate was 1.468 ± 0.248 [6].

UPEC has type 1 pili, which are composed of Fim proteins with FimH at the tip of the fibrillum. The FimH enables it to bind to mannosylated host proteins on the surface of the bladder urothelium [1,7,8]. If a sufficient concentration of a FimH-binding substance is excreted in the urine, it can saturate the FimH adhesin and prevent UPEC type 1 pili from adhering to urothelial glycoprotein receptors. Therefore, FimH inhibitors are anticipated to be a promising therapeutic target for future treatment approaches. Various FimH inhibitors have been synthesized, and their K_D_ values to FimH have been investigated [9]. However, the amounts of most of these synthetic compounds that reach the urinary tract after oral administration in humans have not been investigated. The functions required of a preventive agent against urinary tract infections are that it is safe for humans, that it reaches the urinary tract when administered orally, and that its concentration in the urine is high enough to saturate FimH adhesion.

1-Deoxymannose (DM), also called 1,5-anhydro-D-manntitol, is the C1-deoxyform of Man; therefore, DM is a Man analog [10]. DM is synthesized by the hydrogenation of 1,5-anhydro-D-fructose (1,5-AF) produced from α-glucan, with Pd/C as a catalyst, or *Saccharomyces cerevisiae*, and can enzymatically reduce 1,5-AF in medium and release DM and 1,5-anhydro-D-glucitol, a C2-epimer of DM, in medium [11,12]. The K_D_ value of DM for FimH in vitro has already been reported [13]. The values for DM are lower than those for Man; therefore, DM is expected to be a FimH inhibitor [9]. To evaluate the functionality of drugs or supplements for the prevention of urinary tract infections, the first step involves obtaining experimental data on the urinary excretion rate following oral ingestion in humans. However, such human trials have not been conducted, and it remains unclear whether DM acts as a FimH inhibitor exhibiting activity within the human urinary tract.

Here, we prepared crystals of DM samples for human trials, measured the urinary concentration after oral ingestion of DM, and compared it with the K_D_ value of DM to evaluate its potential as an orally administered FimH inhibitor.

## 2. Results

### 2.1. Preparation of Test Material and Purities

A 1,5-AF aqueous solution was prepared from starch by enzymatic degradation with a purity of 94.8%. Seventy percent of the 1,5-AF was converted to DM after hydrogenation [11,12]. The purity of the collected crystals of DM was determined to be 98.2% using high-performance liquid chromatography (HPLC), which was sufficient for use in a human trial. The purity of the Man supplements was 100%.

As a result of analysis of X-ray diffraction data of DM, the crystal was orthorhombic (rhombic), with the space group P212121 (19) and the following lattice constants: a = 7.906 Å, b = 9.460 Å, c = 9.918 Å, α, β, γ = 90°. The X-ray diffraction profile is shown in Figure 1, and the peak search results are presented in Appendix A (Table A1). When observed under an optical microscope, the crystals are orthorhombic (rhombic) and octahedral (Figure 2).

### 2.2. Safety Analysis of DM

In a single oral gavage administration study of DM in mice, no deaths or abnormalities in general condition were observed in the DM-administered group, and no differences were observed between the control and DM-administered groups in body weight measurements 14 days after administration. Furthermore, no abnormalities were observed in the animals during necropsy. Based on the above results, it was determined that the LD_50_ value of DM for female mice exceeds 2000 mg/kg.

To set the test dose on bacterial reverse mutation test, a range-finding test was conducted using a total of five doses of the test article, 1250, 313, 78.1, and 19.5 μg/plate, which were divided by the common ratio of 4, with 5000 μg/plate being the highest dose. As a result, since no increase in the number of revertant colonies was detected, and growth inhibition and precipitation were observed, this main test was conducted using five doses, of 5000, 2500, 1250, 625, and 313 μg/plate, for all strains, regardless of the presence or absence of metabolic activation. No increase in the number of revertant colonies was observed in any of the test strains, and neither growth inhibition nor precipitation were observed. Based on these results, DM was considered negative.

### 2.3. Human Trial

Six subjects ingested a single dose of 1 g of Man dissolved in water, and we attempted to detect Man concentrations in the urine. These values were below the detection limit of HPLC (detection limit 0.1 mg/mL). Therefore, the determination was performed using gas chromatography–mass spectrometry. Figure 3 shows the individual changes before and after administration. Even before ingestion, the urine samples of five subjects contained from 1.22 μg/mL to 11.3 μg/mL Man, and in one subject this measured 42.5 μg/mL. The average value was 11.4 ± 15.7 μg/mL. After ingestion, one subject showed a decrease in urinary Man concentration, while the concentration in the other subjects remained unchanged. There was no statistically significant increase in urinary Man concentrations after ingestion (1, 2, 3, 4, 6, and 8 h after) compared to before ingestion (*p* = 0.13, 0.09, 0.09, 0.10, 0.34, and 0.14).

One week after the mannose trial, the same subjects ingested a single dose of 1 g DM dissolved in water; however, no abnormal symptoms, such as diarrhea, were observed. Figure 4 shows individual changes in urine DM concentration after ingestion. The DM concentration in the urine of all the participants was below the HPLC detection limit before ingestion. After ingestion, all DM content measurements in the urine increased. In four out of six subjects, the peak concentrations in urine were detected at 2 h and ranged from 1.22 to 5.78 mg/mL. In the remaining two subjects, the peak values were 1.46 mg/mL and 0.665 mg/mL at the 3rd and 6th hour. All urinary DM concentrations after ingestion (1, 2, 3, 4, 6, and 8 h after) were statistically significantly increased compared to before ingestion (*p* = 0.05, 0.03, 0.008, 0.002, 0.005, and 0.005). Furthermore, the DM concentration at all times after ingestion (1, 2, 3, 4, 6, and 8 h after) was significantly higher than that of Man (*p* = 0.05, 0.03, 0.008, 0.002, 0.005, and 0.005).

The urinary recovery rate of DM was 43.3 ± 9.05%. These results indicate that most of the orally injected DM was quickly excreted in the urine.

### 2.4. Utilization of Carbohydrates by E. coli

We evaluated the ability of *E. coli* (NBRC 3301) to utilize DM. The turbidity results are illustrated in Figure 5. No increase in turbidity was observed in the culture solution containing DM compared to that in the culture solution containing Man and glucose. It is difficult to use DM against *E. coli*.

## 3. Discussion

The LD_50_ of DM was more than 2000 mg/kg, and the bacterial reverse mutation test produced negative results, which are revealed in this study. Therefore, DM can be regarded as safe for a single oral administration. On the contrary, in order to conduct long-term human trials on supplements, it is necessary to conduct a 90 d continuous oral administration test and micronucleus tests to ensure safety.

Table 1 shows the K_D_ values of FimH for mannose and DM, which were previously reported and measured using isothermal titration calorimetry, along with the urinary concentrations of DM and Man after oral intake obtained in this study [9,13].

In this Man trial, the average urinary Man concentration before ingestion was 11.4 ± 15.7 μg/mL, which is higher than the K_D_ value of 0.301 μg/mL, so it may be thought that Ⅿan is constantly excreted in the urine, and that Ⅿan has an inhibitory effect on FimH, and may have a function in UPEC. On the other hand, urinary Man concentration did not increase even when 1 g of Man was administered orally. A similar report was made in 2023, and it was reported that upon 2 g oral intake of Ⅿan, there were non-responders whose urinary Man/creatinine ratio did not change, and even in responders, the concentration increased by only about 40% [6]. For these reasons, it is difficult to expect that even if Man is used as a supplement, its effect on the inhibition of UPEC adhesion will be enhanced. In 2024, a large-scale human trial was conducted on the effectiveness of Man in preventing urinary tract infections, and concluded that Man did not have any effect on preventing urinary tract infections compared with a placebo [5]. Man in the urinary tracts of both test groups might have contributed to the prevention of UTIs; however, there may have been no difference in its effectiveness between the groups.

After oral intake of DM, the concentrations of DM in urine rapidly increased dramatically, and the concentration was 3600–31,200 times higher than that of K_D_, and the values were quite large compared with those of Man. One gram of DM is a realistic amount for supplementation.

This study involved six healthy individuals, both men and women, aged between 33 and 57 years. However, due to the limited sample size, restricted age range, and uneven gender ratio, caution is advised when generalizing the findings. It is essential to evaluate the applicability of the results to groups with different age ranges and gender distributions.

Although the sequence identity is only 15% with *E. coli*, type 1 fimbrial FimH of *Salmonella enterica* has the ability to mediate mannose-sensitive adhesion like *E. coli* [14]. Viruses and microorganisms that induce red blood cell agglutination have long been studied. The inhibitory activities of Man and DM have been evaluated in guinea pig red blood cells and *Salmonella Shigella flexneri* [15]. Man and DM inhibit agglutination at 0.025% (250 μg/mL) and 0.013% (130 μg/mL) or higher when tested at 30 times the minimum bacterial concentration to cause hemagglutination. At a bacterial concentration of 1.9 times this value, those of Man and DM were both greater than 0.0016% (16 μg/mL). Comparing these values to the urinary concentrations from our human studies, DM is much higher than the 130 μg/mL of the 30 times test for *Salmonella*, and Man is much lower than 250 μg/mL. At a bacterial concentration of 1.9 times, the concentration of Man was close to that of Man, which is constantly excreted in the urine. From these facts, it is assumed that the constantly excreted Man, although weak, slightly suppresses the adhesion of UPEC with type 1 fimbriae, but cannot suppress adhesion when a large number of bacteria are present. Conversely, we hypothesize that continuous supplementation with DM may effectively suppress infections, even in cases of high bacterial counts. To validate this hypothesis, pharmacokinetic studies involving a larger number of subjects, as well as studies on the prevention of human urinary tract infections, are required.

Although the relationship has not been clearly proven, when urinary glucose occurs in patients with diabetes, sugar may increase the chances of developing a UTI [16,17]. Therefore, the carbohydrates in urine should not be utilized for growth by *E. coli*. Since DM is not assimilated by *E. coli*, it is assumed that the risk of bacterial contamination is low, even when used as a preventive supplement for urinary tract infections over a long period.

In human studies, the urinary recovery of DM was less than 50% after ingestion to 8 h. Therefore, the remainder may have reached the large intestine without being absorbed by the small intestine. UPEC in the intestine reaches the urinary tract via the anus and vagina [1]. Oral mannoside intake in mice reduces UPEC in the intestine and urinary tract [18]. Therefore, pharmacokinetic studies are also needed to determine the amount of DM reaching the intestine.

Alternatives to antibiotics are required to prevent UTIs. Therefore, DM has the potential to act as a highly active prophylactic agent. However, because both FimH and P fimbriae are involved in the adhesion of *E. coli* [1], there is some concern as to whether inhibitors of FimH alone have a preventive effect. In the future, it will be necessary to conduct safety studies, micronucleus tests, and 90 day oral administration test in rats, on DM, and to evaluate its activity in long-term human ingestion prevention studies. Upon ingesting DM, the DM concentration in urine rises rapidly but decreases within a few hours. Therefore, it is crucial to determine an appropriate dosage and interval to maintain its effectiveness.

## 4. Materials and Methods

### 4.1. Test Materials

The DM for the human trial was prepared in the following manner. 1,5-AF was prepared as described previously [19]. DM was synthesized by the method of previous reports with some modifications [11]. Aqueous 1,5-AF (15% *w*/*w*) was used with a 20% nickel sponge as a catalyst at 40 °C, for 3 h under H_2_ pressure 0.9 MPa. Approximately 70% *w*/*w* of 1,5-AF was converted to DM in this reaction. The reaction solution removed the catalyst, which was concentrated to 75% *w*/*w*. After the seed crystal of DM was added, the solution temperature was maintained at 40 °C for 12 h with stirring to crystallize DM. DM crystals were obtained by removing the mother liquor with basket-type centrifugation.

For Man, we used pure powder commercially available for supplements (NOW Health Group, Inc., Bloomingdale, IL, USA).

### 4.2. X-Ray Crystal Structure Analysis of DM

We commissioned CLEARISE Co., Ltd. to perform the X-ray diffraction analysis of the DM crystal. The measurement device used was wide-angle X-ray diffraction device, RINT2500HL (Rigaku Holdings Co., Tokyo, Japan), and the measurement was carried out under the following conditions: measurement wavelength—CuKα (0.15418 nm); X-ray output—50 kV-250 mA; optical system—concentrated beam with monochromator; slit—DS 0.5 deg + 5 mmH; SS 0.5 deg; RS 0.15 mm; scanning axis—2θ/θ interlocking; scanning method—continuous scanning; scanning range—2 ≦ 2θ ≦ 80 deg; scanning speed—0.5 deg/min; sampling—0.01 deg.

### 4.3. Urinary DM Concentrations

Urinary DM concentrations in human trials were measured by high-performance liquid chromatography (HPLC). HPLC analysis was conducted using two MITSUBISHI MCI GEL CK 08S (Mitsubishi Chemical Co., Tokyo, Japan) connection columns (8.0 × 500 mm) with water as the mobile phase. The flow rate was set at 1.0 mL/min, and detection was performed using a differential refractometer detector. The column temperature was maintained at 60 °C, and the injection volume was 100 µL. The total runtime of the analysis was 40 min. The concentration of DM in urine was determined by analyzing the HPLC chromatogram of urine samples using the simple area percentage method. The area value of DM was calculated and then derived based on the ratio to the area value of the DM standard substance.

### 4.4. Urinary Man Concentrations

Urinary Man concentrations in human trials were measured by Gas chromatography–mass spectrometer (GC-MS) analysis. In total, 0.15 micro gram of adonitol was added as an internal standard to a 30 μL urine sample that was diluted 10 or 100 times, and 60 μL ethanol was added to remove the protein. After centrifuging at 15,000 rpm for 5 min, 60 μL of supernatant was dried with a centrifugal evaporator. After adding 20 μL of TMS-PZ (Tokyo Chemical Industry Co., Ltd., Tokyo, Japan) to the dried sample, and allowing it to react for one hour at room temperature, the concentration of Man was measured by GC-MS. The measurements were performed using a TRACE 1610 gas chromatograph-equipped TSQ 9610 mass spectrometer Thermo Fisher Scientific Ink. Waltham, MA, USA) and TG-5SILMS column (Length: 20 m, I.D.: 0.18 mm, Film: 0.18 μm, Thermo Fisher Scientific Ink., Waltham, MA, USA). Helium was used as the carrier gas, and the flow rate was maintained at 0.8 mL/min. One micro liter sample was injected at a temperature of 250 °C, and the split ratio was 1:20. The column oven temperature was kept at 80 °C for 2 min, and then was gradually increased to 300 °C at 20 °C/min. The ion source temperature was 250 °C, and the mass spectra range was between 50 and 550 *m*/*z*.

### 4.5. Acute Oral Toxicity Test of DM Using Female Mice

This study was outsourced to Japan Food Research Laboratories and approved by the Animal Experiment Ethics Committee, Tama Research Institute (TM21081001). The DM dose was 2000 mg/kg, and a control group was set up to receive water for injection, and each group consisted of 5 ICR female mice. The test animals were fasted for 4 h prior to administration. After measuring the body weight, the test group received the DM solution, and the control group received the injection solution in a single dose of 20 mL/kg using a gastric tube. The observation period was 14 d, with frequent observations on the day of administration and once daily thereafter. Body weight was measured on days 7 and 14 after administration. All the animals were necropsied at the end of the observation period.

### 4.6. Bacterial Reverse Mutation Test of DM

The bacterial reverse mutation test was outsourced to Bozo Research Center Co., Ltd., and tested in accordance with good laboratory practice. The protocol followed the OECD Guidelines for Testing of Chemicals 471 for bacterial reverse mutation tests. The strains used in the test were Salmonella typhimurium TA100, TA1535, TA98, and TA1537, as well as *E. coli* WP2 uvrA. 2-(2-Furyl)-3-(5-nitro-2-furyl) acrylamide, sodium azide, CR-191, 2-aminoanthracene, and benzo [a] pyrene were used as positive controls.

### 4.7. Humane Trial

This study was conducted in accordance with the Declaration of Helsinki and approved by the Ethics Committee of User Life Science Co., Ltd. (US L202203). Before the subjects participated in this study, the study director sufficiently explained the purpose and content of this study. After confirming that the subjects fully understood and agreed with the content, they provided their voluntary written consent to participate in this study. This study was conducted on six healthy men and women aged 33–57 years.

A sample for oral intake was prepared by dissolving 1 g Man or DM in water to obtain a 20% *w*/*w* solution. In human studies, urine is collected before sample ingestion. The entire sample was ingested orally at 9:00, and the entire amount of urine was collected after 1, 2, 3, 4, 6, and 8 h. The subjects were allowed to eat breakfast before 9:00 a.m. and lunch at 12:00 p.m. during the test. Those meals did not contain mannose or any components that produce mannose during digestion. After the urine volume was measured, the collected urine was stored in a freezer at −25 °C until measurement. After thawing, Man concentration was measured by GC-MS for Man administration, and DM concentration was measured by HPLC for DM administration. The urinary recovery rate of each sample was calculated from the collected urine volume and urinary DM or Man concentration, and was divided by the administered dose to obtain the urinary recovery rate (%).

### 4.8. Carbohydrate Utilization Test

*Escherichia coli* (NBRC 3301) was shaken at 37 °C for 17 h using 2 mL of M9 minimal medium (Difco™ M9 Minimal Salt supplemented with MgSO_4_ and CaCl_2_, Becton, Dickinson and Company, Franklin Lakes, NJ, USA) containing 1% glucose to obtain a culture solution. In total, 10 microliters of the above culture solution was added to 2 mL of M9 minimal medium containing 1% of each sugar (DM, Man, glucose), and cultured by shaking at 37 °C. During the culture, 100 μL was taken out over time, and the turbidity (600 nm) was measured using a cell with an optical path length of 1 cm.

### 4.9. Statistical Analysis

We calculated the mean, standard deviation, and correlation coefficient and performed a T-test for the measured values of two groups using Microsoft^®^ Excel^®^ 2016. A *p*-value of less than 0.05 was considered to indicate statistical significance.

## 5. Conclusions

DM demonstrates binding affinity for FimH. After oral intake of DM, the concentrations of DM in urine rapidly increased dramatically, and the concentration was 3600–31,200 times higher than that of K_D_, and the values were quite large compared with those of Man. Upon oral ingestion, approximately 50% of the ingested DM is excreted into the urinary tract. These findings demonstrated the potential of DM as an effective oral preventive agent against urinary tract infections in humans. DM has never been used as a supplement. It is necessary to conduct a 90 d continuous oral administration test and micronucleus testing to conduct long-term human trials. After confirming its safety, it is necessary to evaluate the preventive effects against UTI.

## 6. Patents

This article is contingent upon data from two patents: “Urinary tract infection prevention/treatment agent containing 1-deoxymannose” (WO/2024/034495) and “1-deoxymannose crystals precipitated from an aqueous solvent” (Unpublished Japanese patents).

## Figures and Tables

**Figure 1 antibiotics-14-00498-f001:**
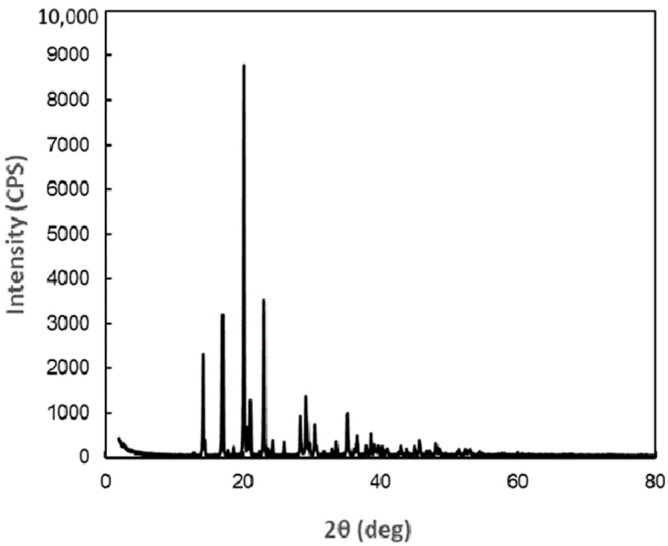
X-ray diffraction profile of 1-deoxymannose crystals.

**Figure 2 antibiotics-14-00498-f002:**
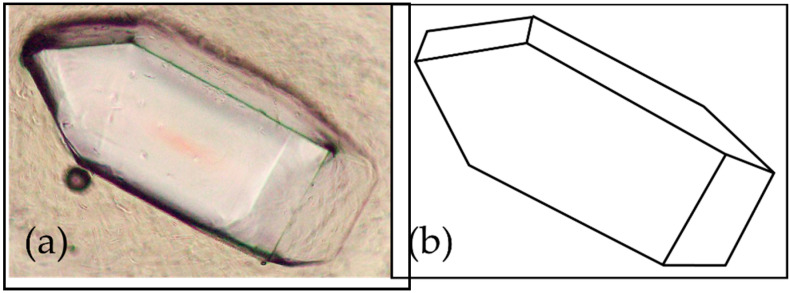
Crystal form of 1-deoxymannose. (**a**) Optical micrograph. (**b**) Model of crystal form.

**Figure 3 antibiotics-14-00498-f003:**
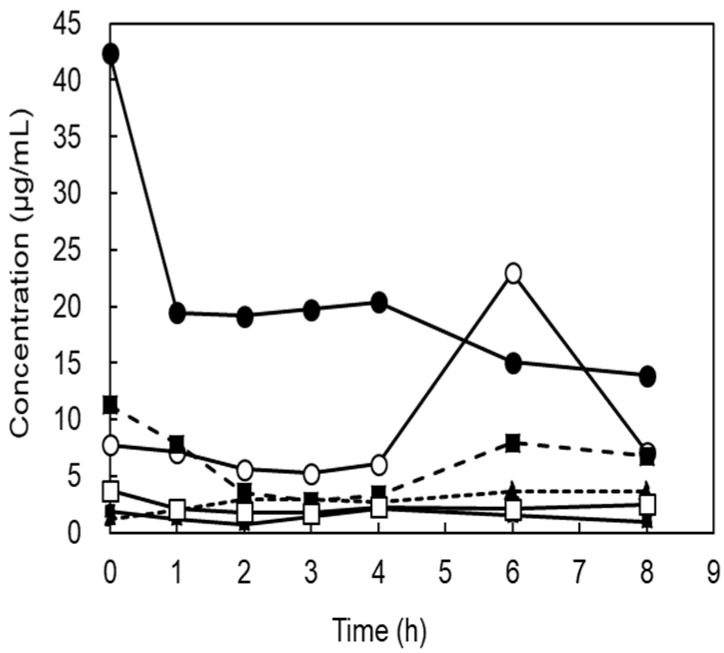
Changes in D-mannose concentration in urine after oral administration. This shows the changes in urinary D-mannose concentration before and after 1 g of D-mannose ingestion for each of the six subjects.

**Figure 4 antibiotics-14-00498-f004:**
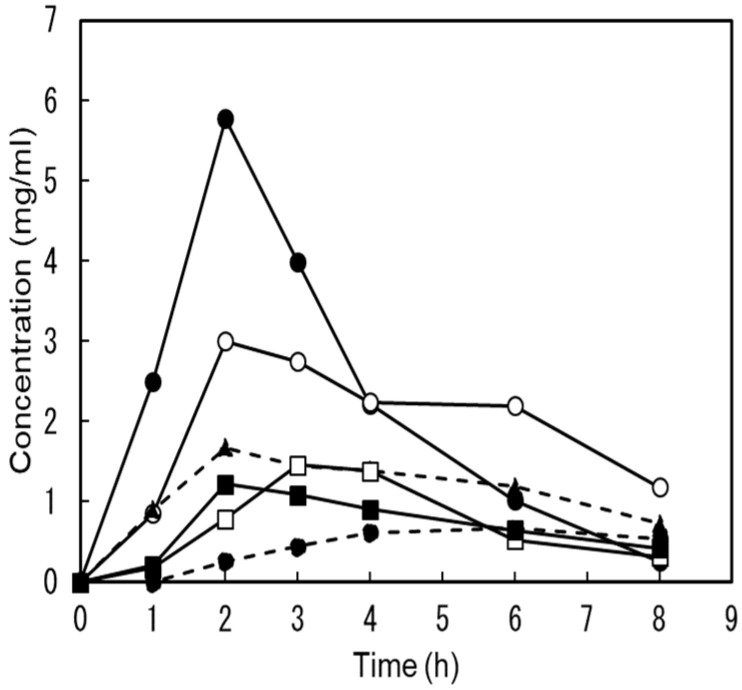
Changes in 1-deoxymannose concentration in urine after oral administration. This shows the changes in urinary 1-deoxymannse concentration before and after 1 g of D-mannose ingestion for each of the six subjects.

**Figure 5 antibiotics-14-00498-f005:**
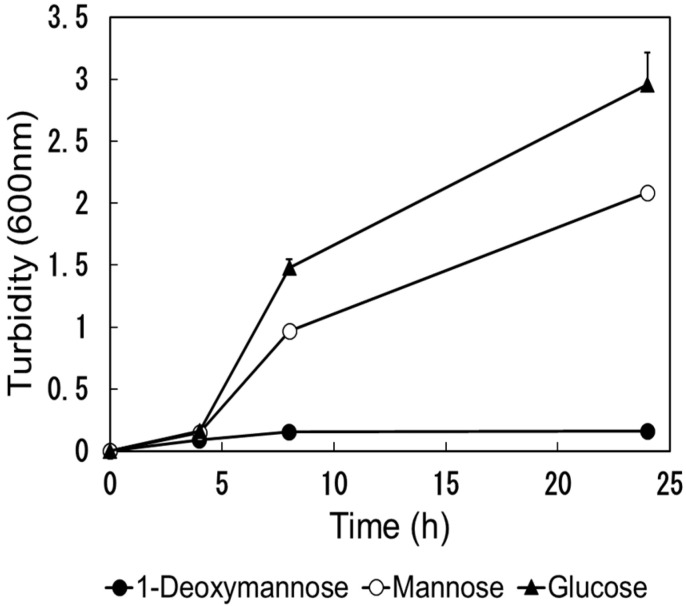
Utilization of 1-deoxymannose on the growth of *Escherichia. coli.* Changes in turbidity (600 nm) caused by the growth of *Escherichia coli* (NBRC 3301) in a M9 minimal medium containing 1% of each sugar (DM, Man, glucose).

**Table 1 antibiotics-14-00498-t001:** Comparisons of K_D_ value to FimH and peak urine concentration.

	1-Deoxymannose	D-Mannose
K_D_ value (µM: µg/mL)	1.125:0.185 *	1.672:0.301 **
Peak urine concentration after oral intake (µg/mL)	665–5780	2.15–22.9
Peak urine concentration/K_D_	3600–31,200	66.3–707
AUC (µg·h/mL)	437–2240	1.43–19.3

* [13], ** [9].

## Data Availability

The original contributions of this study are included in this article. Further inquiries can be directed to the corresponding authors.

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
