# Peer review of "A Pilot Study in Humans on the Urinary Tract Excretion of the FimH Inhibitor 1-Deoxymannose"

_antibiotics, 2025, doi:10.3390/antibiotics14050498_

Round 1

Reviewer 1 Report

Comments and Suggestions for Authors

This article is well written and present interesting topics. Abstract follow the structured abstracting consisting of introduction, objective, methods, results and conclusion. Typo in numbers existed in Abstract. Some comments to be addressed are:

  1. Introduction section is nice, however, I suggest the authors to highlight gap analysis in Introduction.
  2. It is nice if authors add the specific objective in the last Introduction.
  3. The methods should be accompanied with references, if appropriate
  4. During quantification using HPLC, the authors use internal normalization technique, I suggest the authors describe some limitation for this quantification technique.
  5. Since this study involve human trial, it should be accompanied with ethical clearance and the authors have declared "This study was conducted in accordance with the 295
    Declaration of Helsinki and approved by the Ethics Committee of User Life Science"
  6. The discussion related to effect is suggested to be accompanied with statistical test

Author Response

Thank you very much for taking the time to review this manuscript. Please find the detailed responses below.

Comments 1. Introduction section is nice, however, I suggest the authors to highlight gap analysis in Introduction.

Comments 2. It is nice if authors add the specific objective in the last Introduction.

Response 1. Thanks for your comment. I added gap analysis in Lines63-67.

Comments 3. The methods should be accompanied with references, if appropriate.

Response 2. Thanks for your comment. I added reference in Line 235. Other methods were not added, because others were originals.

Comments 4. During quantification using HPLC, the authors use internal normalization technique, I suggest the authors describe some limitation for this quantification technique.

Response 3. I revised methods of HPLC analysis in Lines254-262. If this revise does not satisfy your request, please let us know.  

Comments 5. Since this study involve human trial, it should be accompanied with ethical clearance and the authors have declared "This study was conducted in accordance with the Declaration of Helsinki and approved by the Ethics Committee of User Life Science"

Response 4. I added sentence in Line 297.

Comments 6. The discussion related to effect is suggested to be accompanied with statistical test.

Response 5. I added comments on statistical significance in Lines108-110, Lines119-122 and Lines314-317.

Reviewer 2 Report

Comments and Suggestions for Authors

Dear Editor

Kindly note that my comment on duplication/plagiarized content is empirical only, as I do not have access to professional duplication/plagiarism checking tool(s).

Manuscript title: 1-Deoxymannose: Orally Active FimH Inhibitor for Targeted Therapy in the Human Urinary Tract

Manuscript ID/ file name: antibiotics-3583638-peer-review-v1

Authors: Hayashi et al.

Urinary tract infections (UTI) are continuing health challenge globally. A majority of these infections are due to uro-pathogenic Escherichia coli. These organisms on the other hand are increasing being reported to be antimicrobial resistant. The principal pathogenic mechanism of UPEC include adherence to urinary bladder epithelium through FimH mediated binding and subsequent colonization. Therefore, it would be a rational choice to employ agent like D-Mannose, that prevents binding of Escherichia coli to the bladder wall. While D-mannose has been investigated for its preventive effects, its efficacy in humans remains uncertain.

In this context, the study by Hayashi et al. potential of 1-Deoxymannose (DM) as an orally active inhibitor of FimH. The authors conducted a human trial comparing the urinary concentrations of D-mannose and DM after oral ingestion. While the study is relevant and is important, there are certain weaknesses that need to be addressed by the authors to strengthen the manuscript.

  1. There is authorship change in the pdf document compared to what is there in the online version.

  1. Title appears to be generic and incomplete. The authors should change the title into narrative style so that the content of the manuscript is reflected.
  2. Requires more professional tone of writing for the introduction section.
  3. One major weakness is the number of subjects in the human trial, which is small. Authors admitted this, but they need to show / state with suitable references that similar research design was previously accepted by the scientific community.
  4. In all the sections under materials and methods, each section should start with a purpose statement for the particular experiment being described in the section.
  5. Lines 86-89: As a result ... plate. Unclear phrasing. Please modify.
  6. Line 90: What is S9MIX?
  7. What were the baseline values of mannose in plasma and urine of the subjects?
  8. Fig 3, 4 and 5 are unclear. What do the lines indicate?
  9. Line 128: Was this assertion derived from experimental results? Apparently, there was no LD50 determination experiment? On the otherhand, D-Mannose is generally considered safe and is labeled as GRAS substance.
  10. Lines 132-134: I could not see any isothermal titration calorimetry experiment. Were these data in Table 1 from other previous studies? If so, please mention references.
  11. Lines 185-186: Sentence unclear. Please rephrase.
  12. Lines 190-191: This assertion does not come from the preceding paragraphs (Lines 128-189).
  13. Lines: 198-205: English language needs to substantially upgraded to enable proper reading flow. Moreover, there is no reference in this section.
  14. Lines 207-213: Reference needed.
  15. Lines 215-221: Reference needed. Writing style need to be professional scientific. Authrs may consults similar articles already published in the journal.
  16. Lines 223-235: Reference needed.
  17. Lines 236-245: Was animal ethics clearances obtained? If yes, the same may be mentioned, please.
  18. Line 246-252: Is D-mannose suspected to be mutagenic? If not, the rationale for this experiment needs to be mentioned. Generally, D-Mannose is considered as a GRAS substance as per USFDA.
  19. Lines 263-265: Did the breakfast and lunch for the trial participants contain any D-Mannose? Or any ingredient, which could have been converted into D-Mannone and excreted through urine?
  20. Lines 278-281: Fig 3 mostly shows a falling concentration of DM in urine. If that is correct, then authors need to revisit the assertion made in these lines (278-281).

Comments on the Quality of English Language

English language must be improved.

Author Response

Thank you very much for taking the time to review this manuscript. Please find the detailed responses below

Comment 1. There is authorship change in the pdf document compared to what is there in the online version.

Response 1. Thanks for your comment, but I could not detect the difference in authorship in between PDF and online version. The author in both are “Hiromi Hayashi, Naoto Miyazaki, Takuya Kawakami, Shusaku Izumi, Kazuhiro Yoshinaga”.

Comment 2. Title appears to be generic and incomplete. The authors should change the title into narrative style so that the content of the manuscript is reflected. Requires more professional tone of writing for the introduction section.

Response 2. Thanks for your comment. I revised title to” A pilot study in humans on the urinary tract excretion of FimH inhibitor, 1-Deoxymannose”

Commnet 3. Requires more professional tone of writing for the introduction section.

Response 3. I revised introduction in Lines 45-50.

Comment 4. One major weakness is the number of subjects in the human trial, which is small. Authors admitted this, but they need to show / state with suitable references that similar research design was previously accepted by the scientific community.

Response 4. Thank you for your comment. However, since I could not find any suitable references, I defined this study as a pilot study and commented that a larger study is needed in the future.

           Lines2-3, Line208-212.

Comment 5. In all the sections under materials and methods, each section should start with a purpose statement for the particular experiment being described in the section.

Response 5. I added purpose statement. Lines 233,244,254,264.

Comment 6. Lines 86-89: As a result ... plate. Unclear phrasing. Please modify.

Response 6. I revised sentence as Lines89-93.

Comment 7. Line 90: What is S9MIX?

Response 7. I revised S9MIX to metabolic activation as Lines93.

Comment 8. What were the baseline values of mannose in plasma and urine of the subjects?

Response 8. The mean values ​​of pre-ingestion urinary mannose were added (Line 105-106), whereas plasma was not collected because it would be too invasive for the subjects.

Comment 9. Fig 3, 4 and 5 are unclear. What do the lines indicate?

Response 9. I added explains in Fig 3, 4 and 5.

Comment 10. Line 128: Was this assertion derived from experimental results? Apparently, there was no LD50 determination experiment? On the otherhand, D-Mannose is generally considered safe and is labeled as GRAS substance.

Response 10.  As you suggestion, D-mannose is generally used as food and safe, therefor we did not test LD50 of D-mannose. On the other hand, we evaluated DM safety to conduct human trial.

Comment 11.Lines 132-134: I could not see any isothermal titration calorimetry experiment. Were these data in Table 1 from other previous studies? If so, please mention references.

Response11.I added reference(Lines163-165).

Comment 12.Lines 185-186: Sentence unclear. Please rephrase.

Response12.I corrected sentence (Line 220-221).

Comment 13.Lines 190-191: This assertion does not come from the preceding paragraphs

(Lines 128-189).

Response13.I delate “therefore”.

Comment14.Lines: 198-205: English language needs to substantially upgraded to enable proper reading flow. Moreover, there is no reference in this section.

Response14.I revised methods and added reference(Lines 233-241).

Comment15. Lines 207-213: Reference needed.

Response15.I added reference.

Comment16. Lines 215-221: Reference needed. Writing style need to be professional scientific. Authrs may consults similar articles already published in the journal.

Response16. I revised HPLC method and did not add reference because of HPLC condition was original (Line233-241).

Comment17. Lines 223-235: Reference needed.

Response17. I did not add reference because of GC/MS condition was original.

Comment18. Lines 236-245: Was animal ethics clearances obtained? If yes, the same may be mentioned, please.

Response18. We outsource the animal testing and therefore do not have any animal ethics

clearances.

Comment19. Line 246-252: Is D-mannose suspected to be mutagenic? If not, the rationale

for this experiment needs to be mentioned. Generally, D-Mannose is considered as a GRAS substance as per USFDA.

Response19. As you suggestion, D-mannose is generally used as food and safe, therefor we did not mutagenic test of D-mannose. On the other hand, we evaluated DM safety to conduct human trial. I added DM in Line 278.

Comment20. Lines 263-265: Did the breakfast and lunch for the trial participants contain

any D-Mannose? Or any ingredient, which could have been converted into

D-Mannone and excreted through urine?

Response20. Thanks for your comment. I added sentence to Lines308-309.

Comment21.Lines 278-281: Fig 3 mostly shows a falling concentration of DM in urine. If

that is correct, then authors need to revisit the assertion made in these lines (278-281).

Response21. Fig 3 indicate changes of D-mannose. The concertation of D-mannose of urine may be independent of oral intake.

Reviewer 3 Report

Comments and Suggestions for Authors

This study evaluates the efficacy of orally administered 1-deoxymannose (DM) as a FimH inhibitor and aims to demonstrate its potential as a non-antibiotic prophylactic agent against urinary tract infections by achieving high urinary concentrations post-ingestion. However, a few critical issues should be addressed to ensure the quality of the submission.

1. The human trial is based on only six individuals, which is far too small to draw any statistically meaningful or generalizable conclusions. No power analysis or justification for the sample size is provided, and demographic diversity is lacking.

2. The study focuses solely on pharmacokinetic data (i.e., urinary concentrations of DM) without evaluating whether these levels translate to measurable clinical outcomes, such as reduced bacterial adhesion or UTI incidence in vivo.

3. There is noticeable variability in DM urinary excretion among participants, yet no investigation into potential contributing factors (e.g., metabolic rates, hydration status, gut absorption efficiency) is included.

4. The authors claim DM is a promising prophylactic agent based solely on pharmacokinetics and in vitro KD comparisons, which is a major leap without in vivo efficacy or mechanistic studies in humans or animal infection models.

5. Although an acute toxicity study is included, the paper itself concedes that long-term safety data (e.g., 90-day toxicity, micronucleus assay) are lacking. This significantly limits the practical value of the findings for supplement development.

6. A language service is suggested to improve readability.

Author Response

Thank you very much for taking the time to review this manuscript. Please find the detailed responses below.

Comment 1. The human trial is based on only six individuals, which is far too small to draw any statistically meaningful or generalizable conclusions. No power analysis or justification for the sample size is provided, and demographic diversity is lacking.

Comment 2. The study focuses solely on pharmacokinetic data (i.e., urinary concentrations of DM) without evaluating whether these levels translate to measurable clinical outcomes, such as reduced bacterial adhesion or UTI incidence in vivo.

Comment 3. There is noticeable variability in DM urinary excretion among participants, yet no investigation into potential contributing factors (e.g., metabolic rates, hydration status, gut absorption efficiency) is included.

Comment 4. The authors claim DM is a promising prophylactic agent based solely on pharmacokinetics and in vitro KD comparisons, which is a major leap without in vivo efficacy or mechanistic studies in humans or animal infection models.

Comment 5. Although an acute toxicity study is included, the paper itself concedes that long-term safety data (e.g., 90-day toxicity, micronucleus assay) are lacking. This significantly limits the practical value of the findings for supplement development.

Response. Thanks for your comment. I think all of your comment are suitable. Therefore, I revised as bellow.  

          ・This study was positioned as a pilot study (Lines 2-3) .

          ãƒ»It was noted that there was a statistically significant increase in urinary concentrations of DM after ingestion compared to before ingestion(Line117-120).

          ・It was hypothesized that DM may be effective in preventing urinary tract infections, and it was stated that verification experiments were needed(Lines 119-122).

          ・It was emphasized that safety studies are necessary before functional evaluation studies for DM(Lines 227-230).

Comment 6. A language service is suggested to improve readability.

Response. I submitted this manuscript after having it proofread by a native speaker. It has now been proofread again by another proofreader.

Reviewer 4 Report

Comments and Suggestions for Authors

In the present study, the authors assessed the use of a D-mannose derivative, named 1-deoxymannose (DM), as a potential inhibitor for curing human urinary tract infections. Although, a long-term study (>14 days, e.g. 90 days) will be necessary to fully understand the therapeutic effect of DM, this study provides a good input toward that direction. Therefore, I recommend acceptance after the following points have been addressed:

  1. In the abstract, the following sentence was typed twice: "Six participants were administered single 1-g doses of the samples in a crossover test." (lines 14-15 & lines 17-18). I suggest you remove it from the background section (lines 14-15).
  2. Lines 68-73: Please include references if any.
  3. Figs. 3 & 4: Please add a sentence describing what the six curves in each graph represent.
  4. Line 113: Please fix the following sentence "We evaluated the ability of DM by E. coli (NBRC 3301) was evaluated."
  5. Line 134: The concentration unit should be "µM" not "mM".
  6. Lines 140-142: "A similar report was made in 2023, and it was reported that when 2 g of â…¯an was orally intake, there were non-responders whose urinary Man/creatinine ratio did not change, and even in responders, the concentration increases by only about 40%." Please include the reference.

Author Response

Thank you very much for taking the time to review this manuscript. Please find the detailed responses below.

Comment 1. In the abstract, the following sentence was typed twice: "Six participants were administered single 1-g doses of the samples in a crossover test." (lines 14-15 & lines 17-18). I suggest you remove it from the background section (lines 14-15).

Response 1. Thanks for your comment. I deleted that sentence.

Comment 2. Lines 68-73: Please include references if any.

Response ï¼’.I added the two references on DM conversion(Line 75).

Comment 3. Figs. 3 & 4: Please add a sentence describing what the six curves in each graph represent.

Response 3. I added explains in Fig 3 and 4 .

Comment 4. Line 113: Please fix the following sentence "We evaluated the ability of DM by E. coli (NBRC 3301) was evaluated."

Response 4. I corrected the sentence (Line 127).

Comment 5. Line 134: The concentration unit should be "µM" not "mM".

Response 5. I changed the wording of the sentence based on a suggestion from another reviewer.

Comment 6.Lines 140-142: "A similar report was made in 2023, and it was reported that when 2 g of â…¯an was orally intake, there were non-responders whose urinary Man/creatinine ratio did not change, and even in responders, the concentration increases by only about 40%." Please include the reference.

Response 6. I added a reference (Line 173).

Round 2

Reviewer 1 Report

Comments and Suggestions for Authors

The authors have responded all my concerns, therefore, this paper can be accepted in its present form

Author Response

Thanks for your comment.

Reviewer 2 Report

Comments and Suggestions for Authors

Authors have addressed the comments. However, having animal ethics clearance recorded would add to the strengths of the paper. Editors of the journal need to take a call on this.

My opinion about plagiarisms and self-citations are empirical as I do not have access to professional tools for checking these. Editorial office needs to look into these.

Comments on the Quality of English Language

English language may be improved keeping in view the wider readership of the journal.

Author Response

Thanks for your comment.

 We were able to obtain an animal ethics approval number from our animal testing outsourcer, and this is included in the manuscript.

We have corrected d the p values.